# Addressing Token Uniformity in Transformers via Singular Value Transformation

**Hanqi Yan**[1]         **Lin Gui**[1]         **Wenjie Li**[2]         **Yulan He**[1,3]

[1]Department of Computer Science, University of Warwick, United Kingdom
[2]Department of Computing, The Hong Kong Polytechnic University, China
[3]The Alan Turing Institute, United Kingdom

## Abstract

Token uniformity is commonly observed in transformer-based models, in which different tokens share a large proportion of similar information after going through stacked multiple self-attention layers in a transformer. In this paper, we propose to use the distribution of singular values of outputs of each transformer layer to characterise the phenomenon of token uniformity and empirically illustrate that a less skewed singular value distribution can alleviate the 'token uniformity' problem. Base on our observations, we define several desirable properties of singular value distributions and propose a novel transformation function for updating the singular values. We show that apart from alleviating token uniformity, the transformation function should preserve the local neighbourhood structure in the original embedding space. Our proposed singular value transformation function is applied to a range of transformer-based language models such as BERT, ALBERT, RoBERTa and DistilBERT, and improved performance is observed in semantic textual similarity evaluation and a range of GLUE tasks. Our source code is available at `https://github.com/hanqi-qi/tokenUni.git`.

## 1 INTRODUCTION

In Natural Language Processing (NLP), approaches built on the transformer architecture have achieved the state-of-the-art performance in many tasks [Veitch et al., 2020]. However, recent work identified an anisotropy problem in language representations generated by transformer-based deep models [Ethayarajh, 2019, Gao et al., 2019, Li et al., 2020], i.e., the learned embeddings occupy a narrow cone in the representation space. Such anisotropic shape is very different from what would be expected in an expressive embedding space [Arora et al., 2016, Mu and Viswanath,

2018]. This problem is called *token uniformity* or *information diffusion*, i.e., different tokens share a large proportion of similar information after going through stacked multiple self-attention layers in a transformer. Goyal et al. [2020] showed that using different transformer-encoded tokens in an input sample as a classification unit can achieve almost the same result.

Recently, Dong et al. [2021] found that pure self-attention networks, i.e., transformers without skip-connections and MLPs, have their outputs converging to a rank one matrix, and such rank deficiency can lead to token uniformity. They therefore concluded that skip-connection and MLP help alleviate the token uniformity problem. However, in our experiments, we still observe the token uniformity problem in the full transformer model with self-attention layers, skip-connections and MLPs, even when its output hidden state matrices are full-rank.

In this paper, we instead investigate the token uniformity problem via exploring the distribution of singular values of the transformer-encoded hidden states of input samples. Our analysis reveals that the learned embedding space is a high-dimensional cone-like hypersphere which is bounded by the singular values. Also, skewed probability distribution of singular values is indicative of token uniformity (See in §3.1). Therefore, making the distribution less skewed towards small singular values can help alleviate the token uniformity issue. Unlike existing methods [Gao et al., 2019, Wang et al., 2020] that implicitly or explicitly guide the spectra training of the output embedding matrix by adding a regularisation term to control the singular value decay, we propose a novel approach to address the token uniformity via smoothing the singular value distribution (See in §4.2). In order to verify the effectiveness of our proposed singular value transformation function in transformer-based structures, we apply it to four commonly used large-scale pretrained language models (PTLMs). In particular, the singular value distribution of the final layer output from a PTLM is modified using our proposed transformation function. Then, the transformed singular values are used to re-

*Accepted for the 38th Conference on Uncertainty in Artificial Intelligence* (UAI 2022).

construct the hidden states in the last layer of the PTLM, which are subsequently used for prediction in downstream tasks. Our extensive experiments on a variety of NLP tasks including semantic textual similarity evaluation and a range of General Language Understanding Evaluation (GLUE) tasks [Wang et al., 2019] across thirteen datasets show that our proposed transformation function can effectively reduce the skewness of singular value distributions in qualitative analysis and achieve noticeable performance gains.

Our contribution can be summarised as follows:

- We have presented both geometric interpretation and empirical study of the token uniformity problem. Based on our observations, we have designed a set of desirable properties of singular value distributions and proposed a singular value transformation function to alleviate the token uniformity issue.

- We have proposed a range of methods to evaluate the transformed features in terms of uniformity and the preservation of the local neighbourhood structure.

- Our proposed transformation function has been applied to four widely-used PTLMs and evaluated on both unsupervised and supervised tasks. The results demonstrate the effectiveness of our proposed method on addressing the token uniformity problem while preserving the local neighbourhood structure in the original embedding space.

## 2 RELATED WORK

Transformer-based mask language models, such as BERT [Devlin et al., 2018], ALBERT [Lan et al., 2019], RoBERTa [Liu et al., 2019] and DistilBERT [Sanh et al., 2019], have achieved significant success in NLP. However, token uniformity, i.e., different tokens share similar representations, is commonly observed with the increasing network depth. Many studies [Ganea et al., 2019, Yang et al., 2019] claimed that token uniformity is caused by rank collapse of the layer-wise outputs because the transformer architecture learns the token representation based on the normalised weighted sum of the context representations.

Another line of work, which observed token uniformity in empirical studies, argued that the desirable word representations should be isotropic and focused on studying the distribution of the learned embeddings [Mu and Viswanath, 2018]. Gao et al. [2019] and Bis et al. [2021] defined the problem as *'representation degeneration'* and gave a theoretical analysis, which asserts that this phenomenon is caused by frequencies of rare words. Wang et al. [2020] proposed to add an exponential decay term in training objective so as to control the singular value distribution. All the aforementioned work focused on token-level features and tasks. More recent work argued that the sentence-level features can also

be anisotropic due to the anisotropy in word features. Contrasting learning can also alleviate the anisotropy problem both theoretically and empirically [Carlsson et al., 2021, Gao et al., 2021, Gu and Yeung, 2021]. Li et al. [2020] proposed BERT-flow to transform the representations learned by PTLMs into a standard Gaussian distribution in order to make the token/sentence representations isotropic. Other researchers turned to the whitening-based post-processing strategy to normalise sentence embeddings to derive almost perfectly isotropic representation distribution [Su et al., 2021, Huang et al., 2021].

We argue that on the one hand addressing rank collapse does not necessarily solve the token uniformity problem, as it is still observed even with the network components such as skip-connections which can guarantee the full rank feature space [Dong et al., 2021]. On the other hand, while whitening methods can effectively solving the token uniformity problem, they failed to preserve the local neighbourhood structure of the original embedding space, which is important for downstream tasks. We propose a novel singular value transformation function which can alleviate the token uniformity and at the same time preserve the local neighbourhood structure.

## 3 SINGULAR VALUE DISTRIBUTION OF TRANSFORMER BLOCK OUTPUTS

In a typical transformer block $\ell$, assuming the input token is $x^{l-1}$, the information propagation process is given by:

$$v^l = \text{Self-Attention}(x^{l-1}), \quad \Phi(v^l) = \varphi(W^l v^l + b^l),$$
$$x^l = \text{LayerNorm}(\Phi(v^l) + x^{l-1}) \quad (1)$$

where $\varphi$ is an element-wise nonlinear function applied to a feed-forward layer, whose weight matrix, $W^l \in \mathbb{R}^{n_l \times n_{l-1}}$, transforms the feature dimension from $n_{l-1}$ to $n_l$, Self-Attention$(x^{l-1})$ returns the weighted value vector of all input representations where weights are derived by multiplying the query vector of the current input $x^{l-1}$ with the key vectors from other inputs. Between every two transformer blocks, there is a skip-connection and a layer normalisation. The former mechanism bypasses the transformer block $\ell$ and adds the input $x^{l-1}$ directly to the output $v^l$ of this block, while the latter normalises the input across the feature dimension.

Taking BERT as an example, we assume that the input of the model is $X = x_1 \oplus x_2 \oplus ... \oplus x_m, X \in \mathbb{R}^{n \times m}$, where $x_i \in \mathbb{R}^n$, $m$ is the number of tokens in an input sentence (we do not distinguish special tokens such as [CLS]), and $\oplus$ is the concatenation operator. The output of a transformer block $\ell$ is denoted as $X^l \in \mathbb{R}^{n_l \times m}$, where $n_l$ is the dimension of output feature in layer $\ell$. Without loss of generality, we assume $n_l > m$ for all layers since the embedding size for

tokens is larger than the maximum length of sentences in most BERT models.

Existing work mainly focused on the discussion of the rank of features learned by a transformer-based language model. For example, it was stated in [Dong et al., 2021] that with the growth of depth in a pure transformer model, the rank of the output representation matrix will converge to 1 exponentially. However, in practice, a position embedding matrix, which is usually full rank, is added to the output representation of each layer. In addition, skip connections are used. Therefore, rank collapse rarely occurs. It can be observed from the empirical cumulative density function of singular values from different layers of BERT, derived from real-world data and shown in Figure 2, that there is no zero singular value.

In this section, we instead study the token uniformity problem by the singular value density of the representation matrix $X$ for a deep network with $\ell$ transformer blocks rather than the rank of $X$. Since the distribution of the singular values of $X^l$ determines the extent to which the input signals become distorted or stretched as they propagate through the network, we aim to understand the entire singular value spectrum of $X^l$ in transformers. In particular, we want to study the degree of skewness of the singular value distribution, because highly skewed distributions indicate strong anisotropy in the embedded feature space, the radical reason for token uniformity.

### 3.1 SINGULAR VALUE VANISHING IN TRANSFORMER

In this subsection, we give a geometric interpretation of the problem of vanishing singular values in transformers. We assume that $X^l \in \mathbb{R}^{n_l \times m}$ is a full rank matrix. We can perform SVD on $(X^l)^{\mathsf{T}} = \mathbf{U\Sigma V}$, where $\mathbf{U}$ and $\mathbf{V}$ are orthogonal matrices and $\mathbf{\Sigma}$ is the diagonal singular value matrix. Without loss of generality, we sort the singular values in a descending order, $\lambda_1 \geq \lambda_2 \geq ... \geq \lambda_m \geq 0$, where $\lambda_k, k \in \{1, \cdots, m\}$, is a diagonal element in $\mathbf{\Sigma}$. We can choose a positive value $C$ such that $\lambda_1 \geq ... \geq \lambda_k \geq C \geq \lambda_{k+1} \geq ... \geq 0$, which defines two subspaces, denoted as $\mathcal{S}^l_{[1,k]}$, and $\mathcal{S}^l_{[k+1,m]}$, respectively. For any token embedding, $x \in X^l$, we can find a point $x'$ in the subspace $\mathcal{S}^l_{[1,k]}$ such that their difference is no larger than $C$. That is, we are able to establish the following bound[1]:

**Theorem:** $\forall x \in X^l, \exists x' \in \mathcal{S}^l_{[1,k]}$, where the subspace $\mathcal{S}^l_{[1,k]}$ is defined based on $\lambda_k \geq C \geq \lambda_{k+1}$, then $\|x - x'\|_2 \leq C$.

According to this result, the embedding space is bounded by two components, the largest singular value in the subspace $\mathcal{S}^l_{[1,k]}$, and the upper bound $C$ of the small singular values, as the radius to span the $k$-dimensional $\mathcal{S}^l_{[1,k]}$ subspace

---

[1]The proof is shown in Appendix A.

into the $m$-dimensional space. Furthermore, the weights of the $m$ components in the $\ell$-th layer is constrained by self-attention: $\Sigma_{i=1}^m \alpha_i^l = 1$, which indicates that the embedding space is a spherical cone-like $k$-dimension hypershere. This phenomenon has been observed in many studies [Gao et al., 2019, Zhang et al., 2020, Wang et al., 2020, Li et al., 2020].

We assume the Probability Density Function (PDF) of the distribution of singular values is $f_p(\lambda)$ , where $\lambda$ is the singular values in the learned embedding space, then we can obtain the value of $k$ based on the Cumulative Distribution Function (CDF) of singular values larger than $C$, and the size of input tokens $m$, $k = \lceil \int_C^\infty m \cdot f_p(\lambda) d\lambda \rceil$.

Therefore, the shape of the embedding space is now decided by two parameters: $C$, the radius of the hypercone, and $m - k$, the size of almost vanished dimensions when $C$ is small. However, due to the complex network operations in transformer blocks, it is difficult to derive the exact form of $f_p(\lambda)$. Hence, we resort to empirical study to understand the singular value distribution which appears to be an exponential long-tail distribution, and use the skewness to measure the risk of dimension vanishing in the rest of this paper.

### 3.2 EMPIRICAL STUDY OF TOKEN UNIFORMITY IN BERT

Existing studies have observed the token uniformity issue in PTLMs and skewed singular value distributions of outputs from the intermediate network layers [Goyal et al., 2020]. Few of them though has explored the impact of different shapes of singular value distributions on the downstream task performance. Here, taking BERT as an example, we empirically illustrate that the singular value distribution of outputs from different intermediate transformer blocks on the Corpus of Linguistic Acceptability (CoLA) dataset. The empirical CDF of singular values of the hidden states (i.e., intermediate token representations) from BERT in layer 2, 4, 6, 8, 10 and 12 is shown in Figure 1. It clearly reveals that the CDF is steeper when closer to the origin, which indicates that the probability of singular values $\lambda$ less than a small $x$ is high ($F(x) = Pr(\lambda < x)$).

With the increase of network depth, the CDF curve tends to be steeper, indicating that the shape of the spherical cone-like embedding space tends to be long and narrow, leading to token uniformity. In the last layer for the prediction (i.e., Layer 12), the representation is projected to an embedding space guided by supervised label information, therefore showing a lower degree of token uniformity. Nevertheless, simply leveraging the supervision from labels is not enough to address the problem of vanished dimensions during deep network training.

We calculate the average cosine similarity between every token pair, and [CLS] tokens from different BERT layers as a proxy measure of token uniformity. Figure 2 shows the

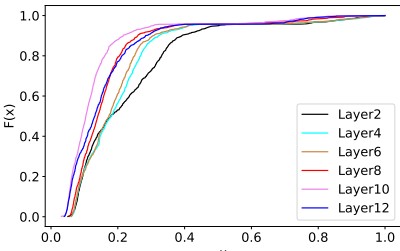

Figure 1: The empirical CDF of singular values from different layers of BERT (on the GLUE-CoLA dataset), $x$-axis: normalised singular values; $y$-axis: CDF of singular values. More flattened curve indicates a more balanced distribution of singular values. For a given $x$, the larger $F(x)$ can cover a higher percentage of singular values which are less than $x$. Hence, the top curve in this figure indicates more vanished dimensions in the embedding space.

skewness of singular values and token uniformity measurement increase as the network layer goes deeper. We also observe the gradual vanishing of smaller singular values as the median of the singular value decreases drastically (from 0.12 to 0.0397). Our results empirically show that the degree of skewness of singular value distributions is indicative of the degree of token uniformity.

# 4 TRANSFORMATION FUNCTION

Having empirically illustrating the changes of the singular value distributions of the transformer layer outputs and the measures of token uniformity across the transformer blocks, we now provide insights of designing a desirable singular value transformation function.

## 4.1 MOTIVATION

As in the geometric interpretation presented in Section 3, the highly skewed singular value distribution in the embedded feature space would mean that the axis of the ellipsoid is significantly longer than the corresponding axis of the sphere, leading to the token uniformity problem. A variety of techniques have been developed to alleviate this problem, and the most popular ones are a series of normalisation methods that can be explained in a unified framework of constraining the contribution of every feature onto a sphere [Ioffe and Szegedy, 2015, Wu and He, 2018, Ba et al., 2016]. A notable example is layer normalisation, an essential module in the Transformer architecture, which scales the variance of an individual activation across the layers to one. One common property of these normalisation methods is that they preserve the trace of the covariance matrix (i.e., the first moment of its singular value distribution), but they do not control higher moments of the distribution, such as the skewness. A consequence is that there may still be a large

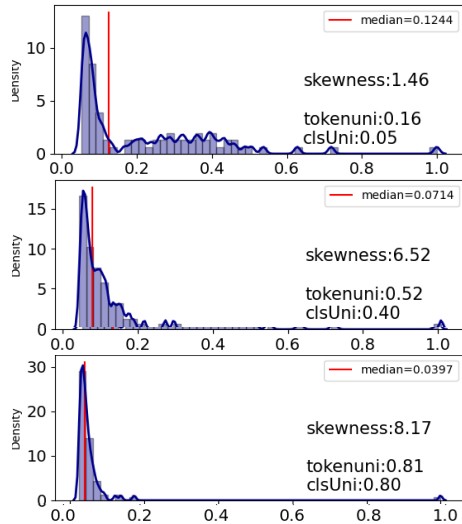

Figure 2: Singular value distribution of the outputs from BERT layer 0, 7 and 12 (from top to bottom) in the GLUE-MRPC dataset. The second moment (skewness), token uniformity and [CLS] uniformity values increase as BERT layer goes deeper, while the median of singular values decreases drastically, close to vanishing.

imbalance in singular values. Here, we propose a transformation function that can adjust the skewness of singular value distributions by modifying small singular values to avoid dimension vanishing.

## 4.2 PROPERTIES OF DESIRABLE SINGULAR VALUE TRANSFORMATION

We want to alleviate the token uniformity problem in PTLMs by adjusting the singular value distributions of outputs of transformer layers (see Section 3.2). Since SVD is computationally expensive[2] and a common practice is to fine-tune a PTLM on downstream tasks, rather than applying the transformation in each transformer block[3], we propose to only apply it in the last transformer block to modify the final output token distribution.

On one hand, we do not want the singular value distribution to be too skewed towards very few large singular values. On the other hand, we do not want it to be too flat as we want to keep the relative difference between singular values learned from powerful pre-trained models. To this end, we propose the following three desirable properties of an singular value transformation function $f(\lambda)$:

---

[2] Approximation methods exist which can speed up the computation of SVD. However, the time cost is still 1.5 times higher than that of the original transformer-based language models.

[3] We have also applied the transformation function to different layers of transformers, but have not observed significant improvements.

**1)** $f'(\lambda) \geq 0$  The large PTLMs have achieved promising performance in a broad spectrum of NLP tasks, showing their capabilities in learning good feature representations. In order to preserve the original data manifold that is mainly defined by the feature singular values, we would like to keep the original order of the singular values. As the input to $f$ is a monotonically increased singular value sequence, $f$ should be monotonically increasing:

$$f(\lambda_i) > f(\lambda_j), \quad \text{if} \quad \lambda_i > \lambda_j, \quad i, j \in [0, n_l - 1].$$

**2)** $f''(\lambda) \leq 0$  To make the transformed singular value distribution more balanced while keeping the largest singular value unchanged, intuitively, we should increase the smaller singular values. The increment $\Delta_i$ for each singular value is defined as $f(\lambda_i) - \lambda_i$ and $\Delta_0 = 0$ (i.e., the largest singular value is kept unchanged). To reduce that gap between larger and smaller singular values, we propose a simple solution that a smaller singular values should have an equal or larger increment than those larger ones while preserving the original order of the singular values. That is:

$$\Delta_i \geq \Delta_j, \quad \text{if} \quad \lambda_i < \lambda_j, \quad \text{where} \quad \Delta_i = f(\lambda_i) - \lambda_i$$

i.e., $\frac{f'(\lambda_i) - f'(\lambda_j)}{\lambda_i - \lambda_j} \leq 0$, the second-order derivative of $f$ should be monotonically decreasing.

**3)** $f(\lambda_{max}) \approx \lambda_{max}$  To guarantee the bounded embedding space, we require the transformation keep the largest singular value unchanged as much as possible. Existing studies have also shown that the largest singular value of the data covariance matrix is significant to model training [Pennington and Worah, 2017].

### 4.3 SOFTDECAY FUNCTION

We develop a non-linear and trainable function built on the soft-exponential function [Godfrey and Gashler, 2015].

$$f(x) = -\frac{\ln(1 - \alpha(x + \alpha))}{\alpha} \quad (2)$$

Specially, when $\alpha < 0$, these curves are monotonically increasing with smaller slop. This is consistent with our properties (1)(2). For the property (3), it can be proved that for any $\lambda$, there is $\lambda \geq f(\lambda)$ when $\alpha < 0$. Hence, we have $\lambda_{max} \geq f(\lambda_{max}) \geq \lambda_2$, where $\lambda_2$ is the second largest singular value.

Combing the desirable properties of singular value distribution and the non-linear transformation function, we describe our proposed transformation in Algorithm 1:

### 4.4 TRANSFORMED FEATURE EVALUATION

Existing research in text representation learning showed that the features should be roughly isotropic (i.e., directionally

---

**Algorithm 1** SoftDecay tranformation
___
**Input:** Original representations $X \in \mathbb{R}^{n_l \times m}$, $m$ is the number of tokens, $n_l$ is the embedding dimension.
1: SVD decomposition $U, \Sigma, V^\intercal = \text{SVD}(X)$
2: Apply transformation $\hat{\Sigma} = \text{SoftDecay}(\Sigma)$
3: Rescaling factor $\mathcal{K} = \max(\lambda)/\max(\hat{\lambda})$
4: Compute transformed singular value $\tilde{\lambda} = \mathcal{K}\hat{\lambda}$
5: Compute transformed representation $\tilde{X} = U\tilde{\Sigma}V^\intercal$
**Output:** Transformed representation $\tilde{X}$
___

uniform) [Mu and Viswanath, 2018, Zhang et al., 2020, Wang et al., 2020, Li et al., 2020, Su et al., 2021] to prevent the feature space squeezing into a narrow cone and preserve as much information of the data as possible. We argue that the evaluation of transformed features should consider both the uniformity and the preservation of local neighbourhood structure in the original embedding space.

**Uniformity.**  We propose to measure the distribution uniformity in three different ways. First, we examine the features similarity (**TokenUni**):

$$\text{TokenUni}(x_i, x_j) = \cos(f(x_i), f(x_j)) \quad (3)$$

where $f(\cdot)$ transforms an input feature by the SoftDecay.

Second, we use the Radial Basis Function (RBF) kernel, **RBF$_{\text{dis}}$**, to measure feature similarity, as it has been shown a great potential in evaluating representation uniformity [Wang and Isola, 2020].

$$\text{RBF}_{dis}(x_i, x_j) = \exp\left(-\frac{\|f(x_i) - f(x_j)\|^2}{t}\right), \quad (4)$$

where $t$ is a constant. We use the logarithmic value of RBF$_{dis}$ in experiments.

Finally, as few predominant singular values will result in an anisotropic embedding space, we can check the difference of variances in different directions or singular values and use the **E**xplained **V**ariance (**EV$_k$**) [Zhou et al., 2021]:

$$\text{EV}_k(f(X)) = \frac{\sum_{i=1}^{k} \lambda_i^2}{\sum_{j=1}^{m} \lambda_j^2}, \quad (5)$$

where $\lambda_i$ is the $i$-th singular value sorted in a descending order, $m$ is the number of all the singular values. In the extreme case when $EV_1$ approximates to 1, most of the variations concentrate on one direction, and the feature space squeezes to a narrow cone.

**Preservation of Local Neighbourhood.**  Ideally, the transformed embeddings should preserve the local neighbourbood structure in the original embedding space. Inspired by the Locally Linear Embedding [Roweis and Saul, 2000], we propose the **L**ocal **S**tructure **D**iscrepancy Score (*LSDS*) to

measure the degree of preserving the original neighbourhood. First, for a data point $x_i$ in the original embedding space, we choose its $k$-nearest-neighbours, then define the weight connecting $x_i$ and its neighbour $x_j$ as the distance measured by the RBF kernel, $w_{ij} = \exp(-\|x_i - x_j\|^2/t)$. In the transformed space, the new feature $\tilde{x}_i = f(x_i)$ is supposed to be close to the linear combination of its original neighbours in the transformed space weighted by the distance computed in the original space:

$$\text{LSDS}(x_i) = \|f(x_i) - \sum_{j \in \mathcal{N}(x_i)} w_{ij} f(x_j)\|^2, \quad (6)$$

where $\mathcal{N}(x_i)$ denotes the $k$-nearest-neighbours of $x_i$.

## 5 EXPERIMENTS

We implement our proposed transformation functions on four transformer-based Pre-Trained Language Models (PTLMs), BERT [Devlin et al., 2018], ALBERT [Lan et al., 2019], RoBERTa [Liu et al., 2019] and DistilBERT [Sanh et al., 2019], and evaluate on semantic textual similarity (STS) datasets and General Language Understanding Evaluation (GLUE) tasks [Wang et al., 2019], including unsupervised and supervised comparison. [4]

### 5.1 UNSUPERVISED EVALUATION ON STS

**Setup** The STS task is a widely-used benchmark of evaluating sentence representation learning. We conduct experiments on seven STS datasets, namely, the SICK-R [Marelli et al., 2014], and the STS tasks (Agirre et al. 2013, 2014, 2015, 2016). We compare our approach with unsupervised methods on adjusting anisotropy in STS tasks, including `BERT-flow` [Li et al., 2020], `SBERT-WK` [Wang and Kuo, 2020], `BERT-whitening` [Su et al., 2021] and `WhiteBERT` [Huang et al., 2021]. `BERT-flow` argued that ideal token/sentence representations should be isotropic and proposed to transform the representations learned by PTLMs into a standard Gaussian distribution. Similar to BERT-flow, `SBERT-WK` also used Natural Language Inference datasets to train the top transformation layer while keeping parameters in the PTLM fixed. `BERT-whitening` and `WhiteBERT` dissect BERT-based word models through geometric analysis on the feature space. Our `SoftDecay` is directly applied to the last layer of the original PTLMs to derive the transformed sentence representation without any fine-tuning [5].

**Results** It can be observed from Table 1 that: (1) whitening-based methods (`BERT-Whitening` and

---

[4]Model training details and additional results can be found in the supplementary material.

[5]Here, we empirically search for the best value of $\alpha$ in $[-0.2, -0.4, -0.6, -0.8, -1.0]$.

`WhiteBERT`), which transform the derived representations to be perfectly isotropic, perform better than the other baselines such as `BERT-flow`, which applies a flow-based approach to generate sentence-embedding from a Gaussian distribution. (2) Our proposed `SoftDecay` gives superior results across all seven datasets significantly, 23.5% of improvement over the base PLTMs and 5% over the best baseline on BERT-based methods. (3) When comparing the results from different PLTMs, we observe more significant improvements on the ALBERT-based models (23%), and modest improvements on the DistilBERT-based models (8%). This is somewhat expected as the token uniformity issue is more likely to occur in deeper models. Therefore, less obvious improvements are found on DistilBERT with only 6 layers, compared to others with 12 layers. The cross-layer parameter sharing in ALBERT could potentially lead to more serious token uniformity, and thus benefits more from the mitigation strategies.

To further understand how `SoftDecay` alleviates token uniformity, we show the CDF of singular values from DistilBERT and ALBERT before and after applying `SoftDecay` in Figure 3. We can observe that before applying `SoftDecay`, the outputs of ALBERT across various layers are very similar while the outputs of DistilBERT across different layers are more different. After applying `SoftDecay`, the singular value distribution of the last layer output (red curve) of ALBERT is less skewed compared to DistilBERT (the brown curve).

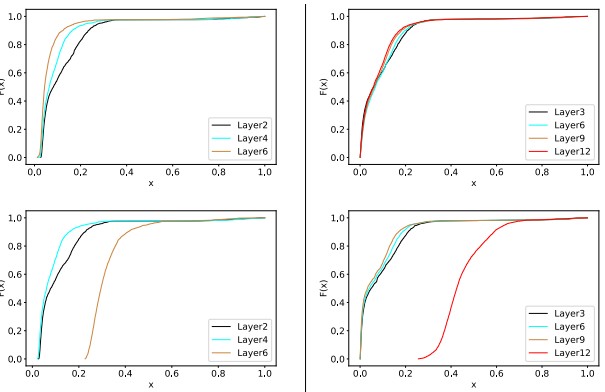

Figure 3: Cumulative distribution function (CDF) of singular values from DistilBERT (left column) and ALBERT (right column), before (top) and after (bottom) applying `SoftDecay` on the STS dataset.

**Feature Evaluation** To gain insights into the characteristics of desirable features for the STS task, we visualise the sentence representations in STS-15 via tSNE and present the results using our proposed metrics in Figure 4. `BERT-Whitening` transforms vanilla features from BERT into perfectly isotropic distribution, which is evidenced in results of the uniformity measures that

| Model | STSB | STS-12 | STS-13 | STS-14 | STS-15 | STS-16 | SICK-R | Avg(Δ%). |
|---|---|---|---|---|---|---|---|---|
| *Results based on Bert-base-cased* | | | | | | | | |
| BERT | 59.05 | 57.72 | 58.38 | 61.97 | 70.28 | 69.63 | 63.75 | 62.97 |
| SBERT-WK [Wang and Kuo, 2020] | 16.07 | 26.66 | 14.74 | 24.32 | 28.84 | 34.32 | 41.54 | 26.64 |
| BERT-flow(NLI) [Li et al., 2020] | 58.56 | 59.54 | 64.69 | 64.66 | 72.92 | 71.84 | 65.44 | 65.38 |
| BERT-whitening(NLI) [Su et al., 2021] | 68.19 | 61.69 | 65.70 | 66.02 | 75.11 | 73.11 | 63.60 | 67.63 |
| BERT-whitening(NLI)-256 [Su et al., 2021] | 67.51 | 61.46 | 66.71 | 66.17 | 74.82 | 72.10 | 64.90 | 67.67 |
| WhiteBERT [Huang et al., 2021] | 68.72 | 62.20 | 68.52 | 67.35 | 74.73 | 72.42 | 60.43 | 67.77(↑7.6) |
| SoftDecay | **72.41**** | **65.16**** | **72.10**** | **69.49**** | **77.09**** | **77.05**** | **65.55**** | **71.26**(↑12.0) |
| *Results based on DistilBERT-base* | | | | | | | | |
| DistilBERT | 61.45 | 59.68 | 59.60 | 63.54 | 70.95 | 69.90 | 63.84 | 64.12 |
| WhiteBERT [Huang et al., 2021] | 69.41 | 61.82 | 66.90 | 67.69 | 74.27 | 72.81 | 59.43 | 67.48(↑5.2) |
| SoftDecay | **71.10**** | **63.33**** | **70.62**** | **68.39**** | **76.34**** | **75.29**** | **63.40**** | **69.78**(↑8.8) |
| *Results based on ALBERT-base* | | | | | | | | |
| ALBERT | 46.18 | 51.02 | 43.94 | 50.79 | 60.83 | 55.35 | 54.99 | 51.87 |
| WhiteBERT [Huang et al., 2021] | 61.76 | 58.33 | 62.89 | 59.92 | 68.84 | 65.90 | 58.03 | 62.24(↑19.9) |
| SoftDecay | **63.30**** | **59.42**** | **62.93**** | **61.09**** | **70.84**** | **68.60**** | **62.26**** | **64.06**(↑23.5) |
| *Results based on RoBERTa-base* | | | | | | | | |
| RoBERTa | 57.54 | 58.56 | 50.37 | 59.62 | 66.64 | 63.21 | 60.75 | 59.53 |
| WhiteBERT [Huang et al., 2021] | 68.18 | 62.21 | 67.13 | 67.63 | 74.78 | 71.43 | 58.80 | 67.17(↑12.83) |
| SoftDecay | **69.47**** | **62.97**** | **67.65**** | **68.09**** | **75.33**** | **73.26**** | **62.87**** | **68.50**(↑15.10) |

Table 1: Spearman's rank results on STS tasks using sentence representation learning methods applied to different PTLMs. Results with ** are significant at $p < 0.001$, * at $p < 0.05$ by comparing with the best baseline. The improvement $\Delta\%$ is calculated by comparing with the base PTLM (first row in each PTLM group).

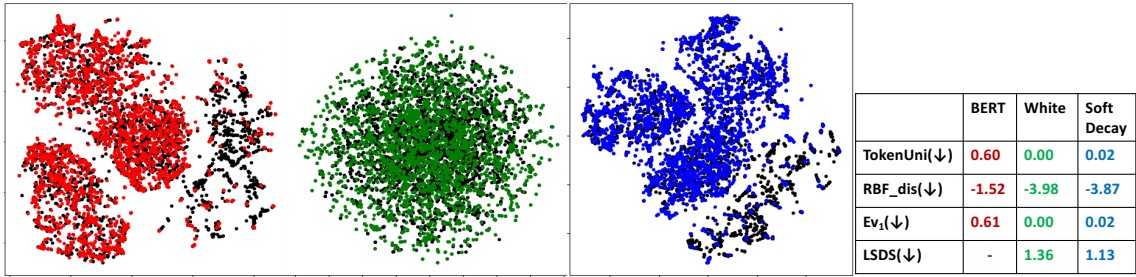

Figure 4: Data points are tSNE mapping results of sentence (pair) representations in STS-15, from left to right derived from the vanilla `BERT`, `BERT+whitening` and `BERT+SoftDecay`. The two sentences in each pair are denoted by two different colors, e.g., black and red in `BERT`. The metrics measuring uniformity and local neighbourhood structure (see in §4.4) are listed on the right. We can see our method preserves the local neighbourhood structure better than `Whitening` with lower *LSDS* and address token uniformity in `BERT` well with lower scores in the first three metrics.

nearly all the features are orthogonal to each other as *TokenUni* is zero and they have the smallest $RBF_{dis}$. It also has the lowest $EV_k$ score of its top singular value. However, `BERT-Whitening` fails to preserve the local neighbourhood of BERT embeddings in its transformed space as shown by its larger Local Structure Discrepancy Score (*LSDS*) compared to `SoftDecay`. By contrast, `SoftDecay` not only significantly improves the uniformity compared to the vanilla BERT feature distribution, but also maintains a similar distribution shape. Our results show that transforming learned text representations into isotropic distributions does not necessarily lead to better performance. Our proposed `SoftDecay` is better in preserving the local neighbourhood structure in the transformed embedding space, leading to superior results compared to others.[6] In Appendix C.3, we further discuss a comparison between `SoftDecay` and a representative contrastive learning method `SimCSE` [Gao et al., 2021], which also aims to alleviate the anisotropy problem in language representations.

## 5.2 SUPERVISED EVALUATION ON GLUE DATASETS

**Setup** We evaluate our method on five sentence-level classification datasets in GLUE [Wang et al., 2019], including grammar acceptability assessment on the Corpus of Lin-

---

[6]The full results of uniformity and structural evaluation of different methods over the seven STS datasets can be found in Appendix C.

| Dataset (size) | BERT | +SoftDecay(Δ%) | ALBERT | +SoftDecay(Δ%) | DistilBERT | +SoftDecay(Δ%) |
|---|---|---|---|---|---|---|
| CoLA(8.5k) | 59.57 | **59.84**\*(↑0.45) | 46.47 | **48.91**\*\*(↑5.25) | 50.60 | **50.73**\*(↑0.26) |
| SST2(67k) | 92.32 | **93.12**\*\*(↑0.87) | **90.02** | 89.91\*(↓0.12) | 90.48 | **91.40**\*\*(↑1.00) |
| MRPC-Acc(3.7k) | 84.00 | **85.20**\*\*(↑1.43) | **85.54** | 85.05(↓0.57) | **84.56** | 84.31\*(↓0.30) |
| MRPC-F1(3.7k) | 89.50 | **89.65**(↑0.17) | **89.67** | 89.28(↓0.43) | **89.16** | 89.00(↓0.18) |
| QNLI(105k) | 91.25 | **91.98**\*\*(↑0.80) | 89.99 | **90.24**\*(↑0.28) | 87.66 | **88.81**\*\*(↑1.31) |
| RTE(2.5k) | 64.98 | **68.23**\*\*(↑5.00) | 66.43 | **68.23**\*\*(↑2.71) | 56.68 | **59.21**\*\*(↑4.46) |

Table 2: Sentence-level classification results on five representative GLUE validation datasets. Matthews correlation is used to evaluate CoLA, Accuracy/F1 is used in other datasets. Δ% represents the relative improvement over the baseline.

| | MNLI | MNLI(mm) | QQP | QNLI | SST2 | COLA | MRPC | RTE | Average(Δ%) |
|---|---|---|---|---|---|---|---|---|---|
| S-BERT | 83.9 | 83.1 | 71.3 | 90.5 | 90.9 | 47.0 | 85.3 | 61.6 | 76.7 |
| BERT-CT | 82.3 | 81.9 | 70.1 | 89.7 | 91.3 | 48.8 | 84.4 | 61.1 | 76.2 |
| SoftDecay | **84.6**\*\* | **84.0**\*\* | **71.6**\* | **90.9**\* | **93.3**\*\* | **50.3**\*\* | **86.2**\*\* | **64.5**\*\* | **78.2 (↑2.6%)** |

Table 3: GLUE test results returned by the GLUE leaderboard. The first two rows are reported in BERT-CT [Carlsson et al., 2021]. Our results outperform BERT-CT by 2.6% on average.

guistic Acceptability (CoLA) [Warstadt et al., 2019], sentiment classification on the Stanford Sentiment Treebank (SST2) [Socher et al., 2013], paraphrase detection on the Microsoft Research Paraphrase Corpus (MRPC) [Dolan and Brockett, 2005], natural language inference on the Question-Answering NLI (QNLI) data and the Recognizing Textual Entailment (RTE) data.[7].

We apply our proposed SoftDecay on top of the last encoder layer in BERT, ALBERT and DistilBERT, and then fine-tune the PTLM weights, along with $\alpha$ on different tasks. In addition to the PTLMs, we include two more baselines, i.e., sentence-level embedding learning models, Sentence-BERT (S-BERT for short) [Reimers and Gurevych, 2019] and BERT-CT [Carlsson et al., 2021] .[8]

- S-BERT adds a pooling operation to the output of BERT to derive a sentence embedding and fine-tunes a siamese BERT network structure on sentence pairs.
- BERT-CT improves the PTLMs by incorporating contrastive loss in the training objective to retain a semantically distinguishable sentence representation.

The two methods aim at making the sentence-level embeddings more discriminative, which in turn alleviate the token uniformity problem.

Since GLUE did not release the test set, the test results can only be obtained by submitting the trained models to the GLUE leaderboard [9]. We show the test results returned by

the GLUE leaderboard in Table 3.

**Results** It can be observed from Table 2 that SoftDecay is more effective on BERT-based model, while gives less noticeable improvement on DistilBERT, similar to what we observed for the STS tasks since DistillBERT has fewer layers. For the vanilla PLTMs, BERT has the better results over all the single-sentence tasks (except for MRPC, sentence-pair paraphrase detection). All the three models achieve better results on inference task (QNLI and RTE), especially on the smaller dataset RTE. The CDF of singular value distributions on RTE before and after applying SoftDecay shown in Figure 5 further verifies the effectiveness of our proposed transformation function. We also observe that models trained on a larger training set tend to generate more similar representations[10]. On MRPC, using SoftDecay is effective on BERT, but gives slight performance drop on ALBERT and DistilBERT. One possible reason is the much smaller training set size. On the GLUE test results shown in Table 3, we observe that SoftDecay outperforms both S-BERT and BERT-CT across all tasks.

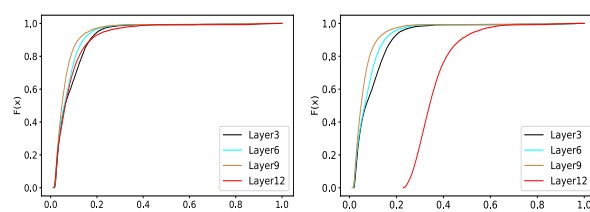

Figure 5: CDF of singular value distributions on RTE before (left) and after (right) applying SOftDecay on BERT. It is clear that SoftDecay can produce a set of larger singular values as evidenced from the curves of $F(x)$.

---

[7]We exclude WNLI as it has only 634 training samples and is often excluded in previous work [Devlin et al., 2018]. We also exclude STS-B as it is a benchmark in the STS task.

[8]We further compare SoftDecay with a method by adding regularisation during training in order to alleviate the anisotropy problem in language representations [Wang et al., 2020] in Appendix D.1.

[9]https://gluebenchmark.com/leaderboard

[10]We investigate the impact of the training set size on model performance in Appendix D.2, Figure 3.

# 6 CONCLUSION AND FUTURE WORK

In this paper, we have empirically shown that the degree of skewness of singular value distributions correlates with the degree of token uniformity. To address the token uniformity problem, we have proposed a singular value transformation function by alleviating the skewness of the singular values. We have also shown that a perfect isotropic feature space fails to capture the local neighborhood information and leads to inferior performance in downstream tasks. Our proposed transformation function has been evaluated on unsupervised and supervised tasks. Experimental results show that our methods can more effectively address token uniformity compared to existing approaches.

Our paper explores the token uniformity issue in information propagation in the transformer encoder, where self-attention is used. It would be interesting to extend our approach to the encoder-decoder structure and explore its performance in language generation tasks. One promising future direction is to improve the generation diversity via addressing the token uniformity since it has been previously shown that anisotropy is related to the word occurrence frequencies [Zhang et al., 2020, Bis et al., 2021]. As such, in the decoding phase, sampling words from more isotropic word embedding distributions could potentially lead to more diverse results.

## Acknowledgements

This work was funded by the the UK Engineering and Physical Sciences Research Council (grant no. EP/T017112/1, EP/V048597/1). YH is supported by a Turing AI Fellowship funded by the UK Research and Innovation (grant no. EP/V020579/1).

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
