# OpenReview forum: "Addressing Token Uniformity in Transformers via Singular Value Transformation"
_auai.org/UAI/2022/Conference — UAI 2022 Poster_

### Official Review · Reviewer_mnp4 · 2022-04-13

**Q2(1) Originality/Novelty:** 3
**Q2(2) Significance/Impact:** 3
**Q2(3) Correctness/Technical Quality:** 3
**Q2(6) Clarity Of Writing:** 3
**Q6 Overall Score:** 7
**Q8 Confidence In Your Score:** 3

**Q1 Summary And Contributions:**

The paper proposes to characterize token uniformity using the
distribution of singular values. It shows empirically that a less skewed singular value distribution
can alleviate the token uniformity problem.  The paper
proposes a novel transformation function for updating the singular
values.  The proposed singular value transformation function is
applied to a range of transformer-based language models. Empirical
evaluation is provided to validate the proposed technique.

**Q2 Assessment Of The Paper:**

More detailed information regarding each of these aspects is given below:

**Q2(5) Reproducibility:**

4: Excellent: Key resources (e.g., proofs, code, data) are available and key details (e.g., proof sketches, experimental setup) are comprehensively described for competent researchers to confidently and easily reproduce the main results.

**Q3 Main Strengths:**

An interesting idea for mitigating token uniformity in transformers.
The proposed technique is validated through experiments.

**Q4 Main Weakness:**

The paper doesn't seem to address the root cause of skewed distribution
of singular values.

**Q5 Detailed Comments To The Authors:**

In the 1st paragraph in Section 3.1, it is unclear the first k token
embeddings correspond to the largest k singular values.
It'd be helpful to show similar plots as in Figure 3, after
SoftDecay transformation.

It's unclear how softdecay is applied, after each layer?
How exactly does softdecay mitigate token uniformity?

There seems to be a typo in Eq 1. There are inconsistent notations,
see the first paragraph under Section 3.1 and Algorithm 1.

The toy example doesn't seem to tell much.

**Q7 Justification For Your Score:**

It is a good paper, addressing an important problem. The main idea is interesting, and validated by experiments.


**Q9 Complying With Reviewing Instructions:**

1: Yes.

---

### Official Review · Reviewer_x984 · 2022-04-13

**Q2(1) Originality/Novelty:** 3
**Q2(2) Significance/Impact:** 3
**Q2(3) Correctness/Technical Quality:** 3
**Q2(6) Clarity Of Writing:** 4
**Q6 Overall Score:** 8
**Q8 Confidence In Your Score:** 4

**Q1 Summary And Contributions:**

This paper focuses on the problem of token uniformity in transformer-based models and first presents a neat geometric interpretation of the problem. The paper shows that the singular values distribution of singular values of outputs of each transformer layer tends have a skewed distribution. It proposes an approach to repair the skewed distribution of singular values with a soft decay transformation while also shining light on the local neighborhood structure & experiments validate the method.

**Q2 Assessment Of The Paper:**

More detailed information regarding each of these aspects is given below:

**Q2(4) Quality Of Experiments (Optional):**

3: Good: The experimental evaluation is adequate, and the results convincingly support the main claims.

**Q2(5) Reproducibility:**

3: Good: Key resources (e.g., proofs, code, data) are available and key details (e.g., proofs, experimental setup) are sufficiently well-described for competent researchers to confidently reproduce the main results.

**Q3 Main Strengths:**

+ The paper contributes to solid theoretical analyses and an important geometric interpretation.
+ The approach is sound and repaid itself is simple and is easy to implement.
+ Excellent experimental results confirms the validity of the approach

**Q4 Main Weakness:**

- I believe an error analysis of the experimental results would be interesting and would complete the paper.
- This is not a strict weakness per se, but I was curious on the utility of the approach for generative tasks - such as machine translation.

**Q5 Detailed Comments To The Authors:**

- It would be very useful to have a section with an analyses of errors - I believe this would be more helpful than the t-SNEs


**Q7 Justification For Your Score:**

+ Neat paper, nicely written, solid theoretical foundations and that shows impressive results.
+ The proposed method is simple to implement

**Q9 Complying With Reviewing Instructions:**

1: Yes.

---

### Official Review · Reviewer_3cqN · 2022-04-14

**Q2(1) Originality/Novelty:** 2
**Q2(2) Significance/Impact:** 2
**Q2(3) Correctness/Technical Quality:** 2
**Q2(6) Clarity Of Writing:** 2
**Q6 Overall Score:** 4
**Q8 Confidence In Your Score:** 4

**Q1 Summary And Contributions:**

This paper have empirically shown that the degree of skewness of singular value distributions correlates with the degree of token uniformity. It also proposed a singular value transformation function by alleviating the skewness of the singular values. They also evaluated proposed transformation function on both unsupervised and supervised tasks. Experimental results show that their methods can more effectively address token uniformity compared to existing approaches.



**Q2 Assessment Of The Paper:**

More detailed information regarding each of these aspects is given below:

**Q2(4) Quality Of Experiments (Optional):**

2: Fair: The experimental evaluation is weak: important baselines are missing, or the results do not adequately support the main claims.

**Q2(5) Reproducibility:**

2: Fair: Key resources (e.g., proofs, code, data) are unavailable but key details (e.g., proof sketches, experimental setup) are sufficiently well-described for an expert to confidently reproduce the main results.

**Q3 Main Strengths:**

1. This paper proposed to use SVD to analyze the token uniformity across layers. Based on this observation, they propose to use  singular value transformation function by alleviating the skewness of the singular values.
2. This paper also evaluated the proposed method in some similarity tasks and some GLUE benchmarks. They show better performance compared to BERT.

**Q4 Main Weakness:**

1. Lack of comparison with other methods such as adding exponential decay term in objective [1].

2. Lack of experiments with other pre-trained language model such as RoBERTa. For sentence eval task, lack of comparison with other methods such as SimCSE [2]. For example, if we combine this methods with simCSE, can we still improve over simCSE.

3. Lack of full results on supervised tasks. The author only report 3 task from GLUE benchmark.

4. The current results is only limited to encoder-style transformer.

[1] Lingxiao Wang, Jing Huang, Kevin Huang, Ziniu Hu, Guangtao Wang, and Quanquan Gu. Improving neural language generation with spectrum control. In 8th International Conference on Learning Representations, ICLR 2020, Addis Ababa, Ethiopia, April 26-30, 2020, 2020.
[2] Gao, Tianyu, Xingcheng Yao, and Danqi Chen. "Simcse: Simple contrastive learning of sentence embeddings." arXiv preprint arXiv:2104.08821 (2021).

**Q5 Detailed Comments To The Authors:**

1. Add missing comparison with other methods.
2. Add experiments on other pre-trained language model such as RoBERTa or T5.
3. report full results on GLUE.
4. extend analysis to transformer cross-attention.

Minor suggestions:
1. add missing citation in first page. There are some question mark in the papers.


**Q7 Justification For Your Score:**

I think the current experimental section is a bit weaker. It will be good if authors can add those missing experiments.

**Q9 Complying With Reviewing Instructions:**

1: Yes.

---

### Decision · Program_Chairs · 2022-05-15

**Decision:**

Accept (Poster)

**Comment:**

Meta Review: This paper studies the token uniformity problem prevalent in transformer-based models (namely, all tokens ending up with similar information in the upper layers of the transformer). It characterizes this phenomenon in terms of the singular value distribution of transformer layer outputs and proposes ways to alleviate the issue by decreasing the skew of singular values.

This is a fairly novel idea that is developed and empirically tested using multiple models and datasets. One reviewer raised important points about additional experiments, which the authors have (to a reasonable extent) addressed in the rebuttal. I would strongly ask the authors to include such additional experiments in their final version.